

# Enhanced Simulation of Supercooled Liquid Water for In-Flight Icing Using an Aerosol-Aware Microphysics Scheme with CAMS Reanalysis

Min Yuan[1,2], Di Wang[1], Weijia Wang[3], Lei Yin[4], Xiaobo Dong[5,6], Delong Zhao[7], Fan Ping[8]

[1]College of Aviation Meteorology, Civil Aviation Flight University of China, Chengdu, China
[2]China Meteorological Administration Key Laboratory for Aviation Meteorology, Chengdu, China
[3]Weather Modification Office of Sichuan Province, Chengdu, China
[4]School of Internet of Things, Nanjing University of Posts and Telecommunications, Nanjing, China
[5]Key Laboratory of Meteorology and Ecological Environment of Hebei Province, Shijiazhuang, China
[6]Weather Modification Center of Hebei Province, Shijiazhuang, China
[7]Weather Modification Center, China Meteorological Administration, Beijing, China
[8]Institute of Atmospheric Physics, Chinese Academy of Sciences, Beijing, China

*Correspondence to*: Min Yuan (yuanm@aliyun.com)

**Abstract.** Aerosol-cloud interactions profoundly influence the properties of supercooled liquid water, which in turn play a critical role in aircraft icing. However, accurately quantifying aerosol emission inventories and their spatiotemporal distributions remains a major challenge. In this study, the Thompson-Eidhammer aerosol-aware microphysics scheme is applied to an in-flight icing event over the high-aerosol-concentration environment of the Sichuan Basin, China. Three numerical experiments with different initial aerosol number concentrations are conducted: Default, Climatology, and Copernicus Atmosphere Monitoring Service reanalysis (CAMS), representing clean and polluted conditions. All three experiments successfully reproduce the synoptic-scale spatial distribution of supercooled liquid water. Compared with the clean environment, the polluted scenarios simulate higher supercooled liquid water mass mixing ratios, greater cloud droplet number concentrations, smaller median volume diameters, and longer cloud system lifetimes. The experiments also reveal that stronger auto-conversion process in clean conditions suppresses supercooled liquid water formation, whereas enhanced accretion process in polluted environments promotes supercooled liquid water depletion. Comparison with in situ aircraft observations indicates that, among the three numerical experiments, the CAMS experiment performs best in capturing high supercooled liquid water contents and large median volume diameters. These findings highlight the importance of real-time aerosol input for improving the simulation of aerosol-cloud interactions and supercooled liquid water characteristics.

## 1 Introduction

In-flight icing is recognized as one of the most hazardous weather phenomena in aviation and continues to pose a significant threat (Cole and Sand, 1991; Petty and Floyd, 2004; Liu et al., 2024). It occurs when an aircraft encounters clouds or precipitation containing supercooled liquid water (SLW), leading to the rapid freezing of droplets upon contact with the




aircraft's surfaces (Potapczuk, 2013; Bromfield et al., 2023). Over the past few decades, extensive in situ aircraft measurements of SLW have been conducted worldwide to support natural icing flight certification, weather modification, and cloud precipitation physics research (Cober and Isaac, 2012; Feng and Zhang, 2014; Dong et al., 2021; Bernstein et al., 2019; Schima et al., 2022; Faber et al., 2024). These observations have been crucial for advancing the understanding of cloud microphysical processes, validating numerical models, and improving in-flight icing forecasts and anti-icing strategies (Lave et al., 2021; Rugg et al., 2023; Zhao et al., 2023).

Aerosol-cloud interaction (ACI), also referred to as the aerosol indirect effect (Twomey, 1977; Rosenfeld et al., 2014; Fan et al., 2016; Yu et al., 2024), plays a critical role in the formation, persistence, and depletion of SLW. On one hand, aerosols act as contact ice nuclei (IN) to initiate ice crystal formation, depleting SLW through riming and the Wegener-Bergeron-Findeisen (WBF) process (Omanovic et al., 2024), commonly referred to as the glaciation effect (Lohmann, 2002). On the other hand, aerosols act as cloud condensation nuclei (CCN), and higher CCN concentrations promote the formation of more numerous but smaller cloud droplets. These smaller droplets are less likely to grow into precipitation through the collision-coalescence process (Rosenfeld, 2000) and are more readily transported to subfreezing altitudes, where they can persist in a supercooled state, thereby enhancing the abundance and lifetime of SLW (Albrecht, 1989). However, when IN concentrations are sufficiently high, even these small supercooled droplets can be efficiently scavenged by ice crystals, further accelerating SLW depletion. In recent decades, with continued industrialization and urbanization, the atmospheric abundance and diversity of aerosol particles have increased (Charlson et al., 1992), rendering the mechanisms of ACI influence on SLW increasingly complex.

An important approach to understanding SLW involves simulating ACI using numerical models coupled with an atmospheric chemistry module (Grell et al., 2005). This coupling dynamically represents key aerosol processes, including emission, transport, turbulent diffusion, chemical transformation (such as the formation of secondary aerosols), as well as wet scavenging and dry deposition. Through real-time treatment of these processes, the models can predict the spatial and temporal evolution of various aerosol species, such as sulfate, nitrate, black carbon, organic carbon, sea salt, and dust, along with their size distributions. Representative implementations of this approach include the MOSAIC aerosol module coupled with WRF-Chem (Zaveri et al., 2008; Chapman et al., 2009) and the Goddard Chemistry Aerosol Radiation and Transport (GOCART) aerosol module (Chin et al., 2000; Ukhov et al., 2020; Collow et al., 2023). However, this methodology comes with the additional limitation of large computational costs and the laborious task of preparing detailed emission inventories for aerosols and gaseous pollutants.

In recent years, numerical models have integrated several double-moment microphysics schemes to simulate ACI, yielding promising advancements in the representation of cloud microphysics (Glotfelty et al., 2019). These schemes typically rely on fixed, prescribed, or simply parameterized aerosol inputs, such as aerosol number concentration, size distribution, and chemical composition, to estimate the number of particles activated as CCN, which subsequently form SLW. Because these schemes do not require complex emission inventories or detailed chemical boundary conditions, they are well suited for proof-of-concept simulations and sensitivity experiments investigating the effects of aerosol perturbations on cloud



microphysics. Notable examples include the Morrison scheme, which supports aerosol activation and ice nuclei parameterization; the WDM6 scheme (Lim et al., 2010), which predicts mass and number concentrations of cloud water, rain, and aerosols; and the widely used Thompson-Eidhammer aerosol-aware scheme. Thompson and Eidhammer (2014) first develop the Thompson-Eidhammer aerosol-aware scheme and use aerosol emissions based on empirical climatological data

to evaluate precipitation in a large winter cyclone. The scheme is then applied to simulate in-flight icing, assessing key characteristics such as liquid water content, median volume diameter (MVD), cloud droplet number concentrations (CDNC), and temperature in aircraft icing environments (Thompson et al., 2017). Weston et al. (2022) apply the scheme to evaluate the influence of different initial CCN number concentrations during two fog events in Namibia. Thomas et al. (2021) use the scheme to successfully capture the spatial and temporal evolution of an extreme rainfall event in southwestern India.

Subsequent studies aim to enhance simulation performance by incorporating more realistic aerosol emissions. Li et al. (2024) use aerosol fields provided by the Common Community Physics Package to improve precipitation simulations over Europe and North America. Wu et al. (2024) improve SLW prediction by integrating aerosol data from the Community Multiscale Air Quality model.

In our previous study of in-flight icing events over the Sichuan Basin, we found that this region exhibits higher aerosol

concentrations, greater CDNC, and smaller MVD compared to other icing environments. We concluded that elevated aerosol levels are one of the key factors contributing to the errors in numerical simulations of SLW (Yuan et al., 2025). To test this hypothesis, the present study employs the Thompson-Eidhammer aerosol-aware microphysics scheme and conducts sensitivity experiments using three different aerosol inputs: the scheme's default setting, climatological mean values, and data from the Copernicus Atmosphere Monitoring Service (CAMS). The focus is on evaluating the scheme's ability to

simulate SLW properties under varying aerosol conditions. This approach aims to improve the simulation of in-flight icing events in high-aerosol environments and to advance our understanding of ACI.

This paper is organized as follows: Section 2 introduces the climatic characteristics of the study area, in situ aircraft observations, aerosol datasets, model configurations, experimental design, and the method used to calculate MVD of SLW, and also provides an overview of the Thompson-Eidhammer aerosol-aware scheme. Section 3 presents the simulated SLW

microphysical characteristics and processes from different experiments, as well as comparisons with observations in the icing region. Section 4 summarizes the main findings of the study.

## 2 Data and methods

### 2.1 Study area

The Sichuan Basin, situated in southwestern China, is a large sedimentary basin bordered by the Tibetan Plateau to the west,

the Qinling Mountains to the north, and the Yunnan-Guizhou Plateau to the southeast. Enclosed by high mountains, the basin floor lies between 200 and 750 m above sea level, while surrounding peaks exceed 4500 m, creating sharp topographic gradients. Annual precipitation ranges from 800 to 1200 mm, primarily concentrated in the summer monsoon season. The



basin's enclosed terrain exerts a pronounced influence on its winter climate: weak winds, limited solar radiation, and high humidity favor the development and persistence of extensive stratiform clouds cover (Jin et al., 2013). Additionally,
topographic confinement facilitates aerosol accumulation during winter pollution episodes, further enhancing persistent cloudiness through ACI (Lu et al., 2023; Lu et al., 2024). This distinctive winter cloud regime not only alters the regional radiative balance and thermal structure but also increases the risk of in-flight icing due to the sustained presence of widespread cloud systems.

## 2.2 In Situ Aircraft Measurements

On December 13, 2015, a Y-12 aircraft conducted a cloud seeding operation for weather modification and in situ cloud observations over the Sichuan Basin. Between 12:00 and 12:30 UTC, the aircraft encountered moderate to severe in-flight icing at an altitude of approximately 3,650 m (Yuan et al., 2025).

In situ cloud measurements was performed using the Droplet Measurement Technologies (DMT) particle detection system onboard the Y-12 aircraft. The system includes the following instruments: the 20 Hz Airborne Integrated Meteorological
Measurement System (AIMMS-20), which provides high-frequency measurements of temperature, humidity, and pressure; the Passive Cavity Aerosol Spectrometer Probe (PCASP), which measures the size and concentration of aerosol particles, typically in the range of 0.1-3 μm; the Cloud Droplet Probe (CDP), which detects cloud droplets with diameters ranging from 2-50 μm, covering the majority of typical cloud droplet sizes; the Cloud Imaging Probe (CIP), which captures images of large cloud particles with diameters from 25-1600 μm; the Precipitation Imaging Probe (PIP), which measures precipitation
particles in the 100-6000 μm range; and the hot-wire liquid water content sensor (LWC-300), which provides estimates of bulk liquid water, although its data are not consistently available and often lack baseline adjustments.

For this study, The CDP served as the primary instrument for analysing SLW properties, ensuring consistency across the study. Uncertainty in cloud droplet measurements derived from the CDP is estimated at ± 10%, primarily due to particle sizing errors and the assumption of spherical droplets (Lance et al., 2010; D'Alessandro et al., 2021). In saturated
environments, the PCASP's heated inlet may not fully evaporate cloud droplets, potentially leading to the detection of oversized pseudo-particles (Kleinman et al., 2012). Furthermore, some aerosol particles can act as CCN and grow into cloud droplets, increasing measurement uncertainty for aerosols inside clouds compared to those measured outside (Yu et al., 2022). Consequently, only aerosol data collected outside clouds were used in this study.

## 2.3 Aerosol datasets

Two aerosol datasets are used in this study. The first dataset is derived from the three-dimensional monthly averaged aerosol climatology file provided by the Weather Research Forecasting (WRF) Preprocessing System (WPS). It is based on simulations from the GOCART model for the period 2001-2007, with a horizontal resolution of 0.5°×1.25° (Ginoux et al., 2001; Colarco et al., 2010). This dataset includes both water-friendly aerosol (WFA) and ice-friendly aerosol (IFA) types.





The second dataset is derived from CAMS global reanalysis of atmospheric composition, which provides near-real-time
aerosol data with a horizontal resolution of 0.75°×0.75° and a temporal resolution of 3 hours (Morcrette et al., 2009;
Benedetti et al., 2009; Inness et al., 2019). In the Thompson-Eidhammer aerosol-aware microphysics scheme, dust aerosols
with particle sizes greater than 0.5 μm are treated as hydrophobic aerosols and can act as IN. In contrast, hydrophilic organic
matter aerosols, hydrophilic black carbon aerosols, sea salt aerosols, and sulfate aerosols are considered CCN. Accordingly,
in the CAMS dataset, dust aerosols represented by the variables aermr05 (0.55-0.9 μm) and aermr06 (0.9-20 μm) are
classified as IN. Sea salt aerosols, represented by aermr01 (0.03-0.5 μm), aermr02 (0.5-5 μm), and aermr03 (5-20 μm), along
with hydrophilic organic matter (aermr08), hydrophilic black carbon (aermr10), and sulfate aerosols (aermr11), are classified
as CCN (see Table 1). Compared to climatological datasets, CAMS offers improved temporal resolution and more accurate
regional aerosol distributions, making it particularly suitable for studies requiring high-resolution, time-varying aerosol
inputs.

Since the aerosol concentrations provided by CAMS are expressed in mass mixing ratios (kg kg$^{-1}$), whereas the Thompson-
Eidhammer aerosol-aware microphysics scheme requires aerosol number concentrations (kg$^{-1}$), a conversion is performed in
this study. For aerosol species without specified particle size ranges, namely, hydrophilic organic matter, hydrophilic black
carbon, and sulfate, aerosol number concentrations are derived using typical particle sizes and densities representative of
East Asian conditions (Chin et al., 2002; Li et al., 2020). For dust and sea salt aerosols with defined particle size intervals,
the median diameter of each size range is used as the representative particle size for the conversion (see Table 1).

**Table 1 Types of aerosol represented in CAMS and their typical particle sizes and densities.**

| Name in CAMS | Property | Composition | Particle size range (um) | Typical particle size (um) | Typical density (kg m$^{-3}$) |
|---|---|---|---|---|---|
| aermr05 | IFA | Dust | 0.55-0.9 | 0.7 | 2500 |
| aermr06 | IFA | Dust | 0.9-20 | 10 | 2500 |
| aermr01 | WFA | Sea salt | 0.03-0.5 | 0.26 | 2200 |
| aermr02 | WFA | Sea salt | 0.5-5 | 2.8 | 2200 |
| aermr03 | WFA | Sea salt | 5-20 | 12 | 2200 |
| aermr08 | WFA | Hydrophilic organic matter | | 0.2 | 1800 |
| aermr10 | WFA | Hydrophilic black carbon | | 0.1 | 1000 |
| aermr11 | WFA | Sulphate | | 0.2 | 1770 |




## 2.4 Model configuration

The WRF model is employed to simulate the in-flight icing event. A two-way nesting approach was used, comprising three
nested domains. All domains are initialized simultaneously and integrated for 24 h, from 00:00 UTC on December 13 to
00:00 UTC on December 14. The model configuration includes 48 vertical levels, with the model top located above 50 hPa
and 17 levels below 5 km altitude. Initial and lateral boundary conditions are provided by the NCEP/FNL reanalysis dataset,
with a horizontal resolution of 1.0°×1.0°. As shown in Fig. 1, the horizontal resolutions of the outermost to innermost
domains are 9 km, 3 km, and 1 km, respectively, with corresponding time steps of 30 s, 10 s, and 3.3 s. The outermost
domain covers most of China and is used to analyse the synoptic-scale patterns and spatial distribution of SLW, while the
innermost domain focuses on the southern Sichuan Basin and is used to investigate the microphysical characteristics of SLW.

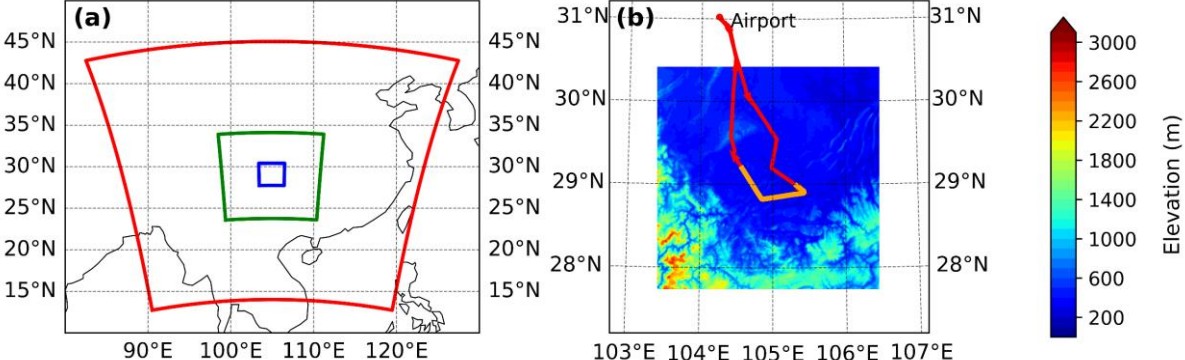

**Figure 1 (a) Configuration of nested domains used in the WRF simulations (red: the outermost domain, green: the second domain, blue: the innermost domain), (b) Topography of innermost domain and flight paths (red), orange represents the location of in-**
**flight icing, arrow represents the direction of flight.**

In addition to the Thompson-Eidhammer aerosol-aware microphysics scheme (Thompson and Eidhammer, 2014), shortwave
and longwave radiation are controlled by the Dudhia scheme (Dudhia, 1989) and the Rapid Radiative Transfer Model
(RRTM) scheme (Mlawer et al., 1997), respectively. The Noah Land Surface Model (Chen and Dudhia, 2001) is employed
to simulate land surface processes, while the Yonsei University scheme is used for planetary boundary layer
parameterization (Hong et al., 2006). The Kain-Fritsch convection parameterization scheme (Kain, 2004) is activated in the
outermost domain but deactivated in the middle and innermost domains, as the higher spatial resolution in these regions
allows for an explicit representation of convective processes (Yu and Lee, 2010).

The Thompson-Eidhammer aerosol-aware microphysics scheme is an advanced and widely adopted double-moment bulk
cloud microphysics scheme implemented in the WRF model. The core of this scheme lies in its ability to explicitly
incorporate the influence of aerosols into the initial formation processes of cloud droplets and ice crystals. The aerosol-aware
capability is achieved through the explicit handling of two key aerosol variables: WFA and IFA number concentration. WFA
refers to hygroscopic aerosol particles that readily activate into cloud droplets. The scheme calculates the number
concentration of activated cloud droplets as a function of WFA number concentration, atmospheric supersaturation, and
temperature, based on a built-in CCN activation parameterization. IFA, on the other hand, represents aerosol particles



capable of acting as IN under supercooled conditions. The number concentration of ice crystals is diagnosed using an IN activation scheme that depends on the IFA number concentration, temperature, and relative humidity. The WFA and IFA number concentrations can be specified through several approaches: fixed background values, offline input from observational or reanalysis datasets, or online coupling with chemical transport models.

## 2.5 Numerical experiments

To examine the impact of initial aerosol conditions on the simulation of SLW, three numerical experiments are designed with different initial aerosol configurations: (a) The default vertical aerosol profile of the Thompson-Eidhammer aerosol-aware microphysics scheme (Default), in which the CCN concentration decreases exponentially from $3*10^2$ cm$^{-3}$ near the surface to 50 cm$^{-3}$ at higher altitudes, and the IN concentration decreases exponentially from 1.5 cm$^{-3}$ near the surface to 0.5 cm$^{-3}$ at higher altitudes, without accounting for the spatiotemporal variability; (b) The three-dimensional monthly averaged

aerosol climatology dataset provided by the WPS (Climatology); (c) Real-time aerosol number concentration data from CAMS on December 13, obtained after converting the original mass mixing ratios to number concentrations. All experiments employ identical dynamical and physical parameterizations, differing only in their initial aerosol fields. The initial and boundary conditions are derived from the NCEP/FNL reanalysis, ensuring consistency in the large-scale environmental background.

To compare the initial CCN and IN number concentrations across different numerical experiments, Figure 2 presents their vertical profiles within the in-flight icing region (indicated by the orange path in Fig. 1b) and compares them with in situ aerosol observations obtained from PCASP. It is important to note that PCASP records total aerosol particle concentrations, without distinguishing between aerosol types. Near the surface (Figure 2a), the default experiment shows a CCN concentration of approximately $2*10^2$ cm$^{-3}$, which is about two orders of magnitude lower than that of the Climatology and

CAMS experiments ($2*10^4$ cm$^{-3}$ and $10^4$ cm$^{-3}$, respectively). This discrepancy reflects the idealized, relatively clean environment represented by the Default experiment. In contrast, the Climatology and CAMS experiments represent more polluted background conditions. Notably, the CAMS experiment aligns more closely with the PCASP observations and more accurately reproduces the vertical distribution of aerosols. For IN (Figure 2b), the CAMS experiment exhibits a concentration on the order of $10^{-3}$ cm$^{-3}$, which is approximately three orders of magnitude lower than the values in the

Climatology and Default experiments, indicating a much lower IN environment.



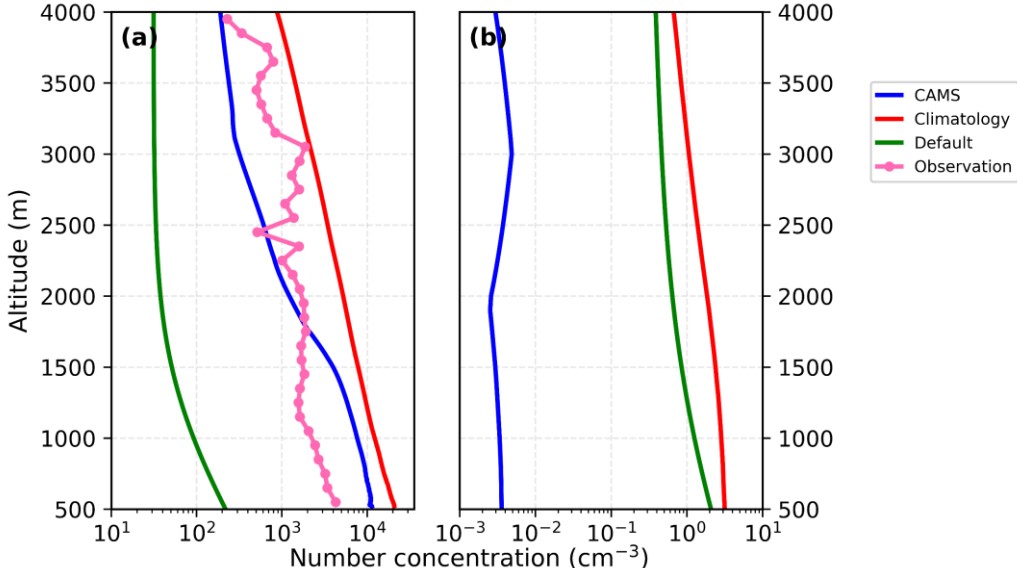

**Figure 2 Vertical profiles of initial CCN (left) and IN (right) number concentrations for Default (green), Climatology (orange), CAMS (blue) experiments, along with the aerosol number concentration from PCASP (pink).**

### 2.6 Prediction of MVD

The Thompson-Eidhammer aerosol-aware double-moment microphysics scheme predicts both the mass mixing ratio and number concentration of cloud water, enabling the diagnosis of characteristic droplet sizes such as the MVD. The scheme represents the cloud droplet size distribution using a generalized gamma distribution (Thompson et al., 2008):

$$N(D) = N_0 D^\mu e^{-\lambda D} \, , \tag{1}$$

where $N(D)$ represents the number of droplets per unit volume, $N_0$ is the intercept parameter, and $\mu$ is the shape parameter of
the size distribution, given by:

$$\mu = min\left(15, \frac{1000}{N_c} + 2\right), \tag{2}$$

where $N_c$ represents the CDNC in units of droplets per cubic centimeter. The slope parameter $\lambda$ in equation (1) is defined as:

$$\lambda = \left[\frac{\pi}{6} \rho_w \frac{\Gamma(4+\mu)}{\Gamma(1+\mu)} \left(\frac{N_c}{LWC}\right)\right]^{\frac{1}{3}}, \tag{3}$$

where $\rho_w$ is the density of water, LWC is the liquid water content. Based on the above equations and descriptions, the MVD
can be diagnosed using the relation derived by:

$$MVD = \left(\frac{3.672 + \mu}{\lambda}\right), \tag{4}$$



## 3 Results

### 3.1 Spatiotemporal distribution of SLW

Figure 3 compares the simulated synoptic-scale distributions of the temperature field and SLW path at 12:00 UTC on 13
December 2015 with ERA5 reanalysis data. Although the aerosol concentrations differ substantially among the three experiments, their simulated spatial distributions of temperature and SLW are generally consistent, except for a notable difference in SLW magnitude (Fig. 3a-c). Specifically, a warm temperature ridge is present between 105°E and 110°E, north of 30°N, with a temperature of approximately −4 °C at the in-flight icing location. The terrain's blocking effect on airflow is evident in the SLW distribution: SLW is primarily concentrated in the Sichuan Basin and its eastern surroundings, between
−6 °C and 0 °C, east of the Tibetan Plateau and north of the Yunnan-Guizhou Plateau. The maximum SLW values occur along the basin margin southwest of the in-flight icing location. The Climatology and CAMS experiments simulate significantly higher SLW values than the Default experiment.

The synoptic-scale distributions of the temperature field and SLW path revealed by the ERA5 reanalysis data (Fig. 3d) are generally similar to those of the three numerical experiments. However, notable differences exist. ERA5 fails to capture the
mesoscale warm ridges located to the east and southwest of the in-flight icing location and simulates the SLW maximum farther east, near 110°E, outside the Sichuan Basin. These discrepancies can be attributed to several factors, including the coarse spatial resolution of ERA5, which limits its ability to resolve small-scale topographic influences, and systematic errors in ERA5's treatment of SLW (Hellmuth et al., 2025). In contrast, the higher spatial resolution of the WRF model allows for a more realistic representation of terrain-induced local circulations and their effects on the thermal and moisture
fields.





**Figure 3 Spatial distribution of SLW path (shaded, unit: kg m$^{-2}$) and temperature fields (colour lines, unit: °C) for (a) Default, (b) Climatology, (c) CAMS, and (d) ERA5 reanalysis.**

Figure 4 presents the time-altitude evolution of hydrometeor properties averaged over the in-flight icing region (orange path in Fig 1b). In the default experiment, both the SLW content and number concentration are relatively small due to the low aerosol concentrations. In contrast, the Climatology and CAMS experiments, which feature higher aerosol concentrations, simulate greater SLW content and number concentration. This enhancement is attributed to the activation of a greater number of CCN into cloud droplets under elevated aerosol conditions, thereby promoting SLW formation, a manifestation of the first aerosol indirect effect (Twomey, 1977). Since SLW is primarily generated during night-time hours (12:00-20:00




UTC), the increased CDNC does not affect cloud albedo or the radiative cooling of the cloud system. This is reflected in the similar heights of the isotherms across all three experiments. In the cleaner environment (Default), SLW is more efficiently depleted through collision-coalescence processes, leading to a reduction in SLW between 3000 and 4000 m beginning around 14:00 UTC and a shortened cloud lifetime. In contrast, in the more polluted environments (Climatology and CAMS), reduced rain formation via collision-coalescence delays the depletion of SLW until approximately 17:00 UTC, thereby

extending the cloud lifetime. This behavior aligns with the second aerosol indirect effect, which associates higher CDNC with suppressed precipitation and prolonged cloud duration (Albrecht, 1989).

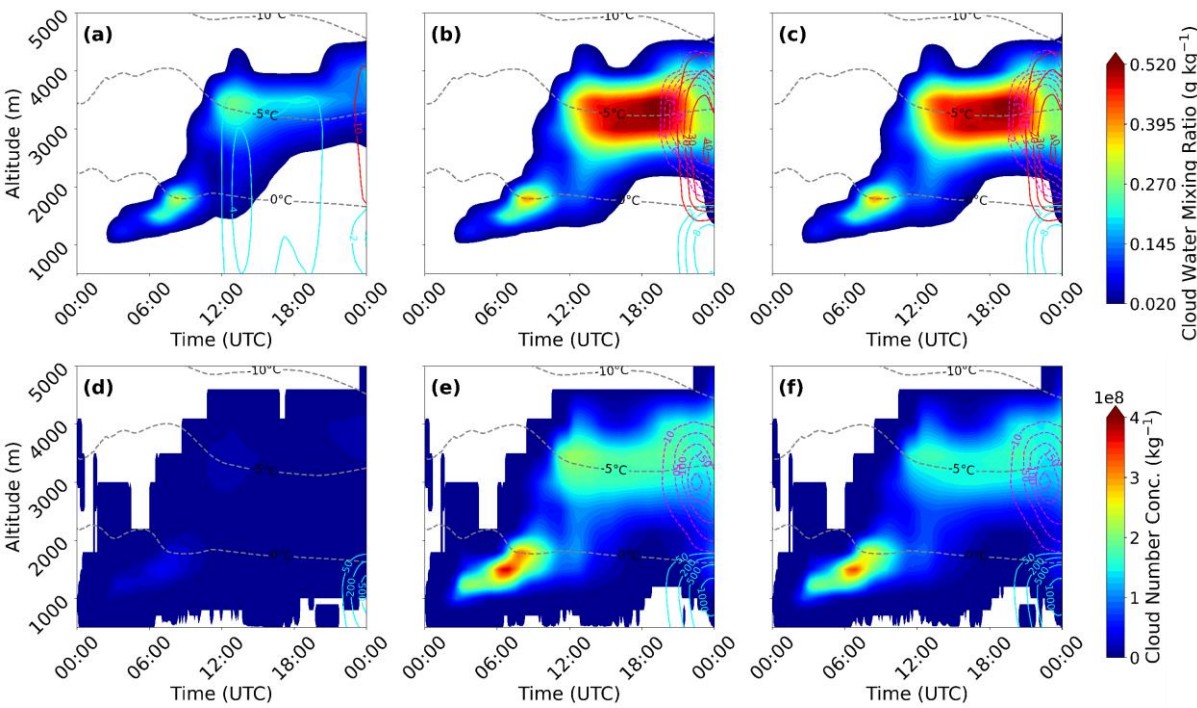

**Figure 4** Domain-average time-altitude cross section of hydrometeor mass mixing ratio (top, cloud water: g kg$^{-1}$, rain: 10$^{-3}$ g kg$^{-1}$,
cloud ice: 10$^{-7}$ g kg$^{-1}$, snow: 10$^{-3}$ g kg$^{-1}$, temperature: °C) and number concentration (bottom, kg$^{-1}$) averaged over the in-flight icing region (orange path in Fig. 1b) for the three experiments: Default (left), Climatology (middle) and CAMS (right), (shading: cloud water, cyan: rain, magenta: cloud ice, red: snow, gray: temperature).

**3.2 SLW microphysics**

To quantitatively evaluate the simulated hydrometeor properties in the in-flight icing region, Table 2 summarizes the
properties of hydrometeors for the three experiments. Compared with the average cloud water mass mixing ratio of 0.178 g kg$^{-1}$ in the Default experiment, both the Climatology and CAMS experiments exhibit higher values. However, the difference between Climatology and CAMS is relatively small (see Fig. 4b-c), with both exhibiting a median value of 0.264 g kg$^{-1}$ and mean values of 0.324 g kg$^{-1}$ and 0.321 g kg$^{-1}$, respectively. In contrast, the number concentrations of cloud water differ more significantly between the two polluted experiments (Fig. 4e-f), with median values of 1.95*10$^8$ kg$^{-3}$ for



Climatology and $1.71*10^8$ kg$^{-3}$ for CAMS. This is attributed to the higher aerosol concentrations in the Climatology experiment.

Based on the simulated cloud water mass mixing ratio and number concentration, the MVD can be diagnosed using the method described by Thompson et al. (2008). In the cleaner environment (Default), the lower CDNC results in a larger MVD, with an average of 24.8 μm, significantly greater than those in the Climatology (13.4 μm) and CAMS (14.1 μm) experiments.

The larger droplet size enhances the efficiency of collision-coalescence processes, leading to greater rain formation (Hoffmann and Feingold, 2023). This is reflected in the higher median rain mass mixing ratio in the Default experiment ($1.36*10^{-3}$ g kg$^{-1}$), which is approximately an order of magnitude greater than those simulated under the more polluted conditions.

**Table 2 The percentiles of the mass mixing ratios, number concentrations, and MVD of hydrometeors for the Default, Climatology,**
**and CAMS experiments.**

| Hydrometeors | Experiments | Percentiles | | | | | |
|---|---|---|---|---|---|---|---|
| | | 1th | 25th | 50th | 75th | 99th | Mean |
| Cloud water mass mixing ratio (g kg$^{-1}$) | Default | $7.47*10^{-2}$ | $7.40*10^{-2}$ | $1.65*10^{-1}$ | $2.56*10^{-1}$ | $5.47*10^{-1}$ | $1.78*10^{-1}$ |
| | Climatology | $3.83*10^{-4}$ | $9.90*10^{-2}$ | $2.64*10^{-1}$ | $5.05*10^{-1}$ | $9.94*10^{-1}$ | $3.24*10^{-1}$ |
| | CAMS | $3.27*10^{-4}$ | $9.84*10^{-2}$ | $2.64*10^{-1}$ | $5.02*10^{-1}$ | $9.78*10^{-1}$ | $3.21*10^{-1}$ |
| Cloud water number concentration (kg$^{-1}$) | Default | $1.80*10^4$ | $1.18*10^7$ | $1.72*10^7$ | $2.58*10^7$ | $7.95*10^7$ | $2.13*10^7$ |
| | Climatology | $1.06*10^7$ | $1.50*10^8$ | $1.95*10^8$ | $2.64*10^8$ | $9.78*10^8$ | $2.38*10^8$ |
| | CAMS | $1.35*10^7$ | $1.21*10^8$ | $1.71*10^8$ | $2.30*10^8$ | $7.39*10^8$ | $1.97*10^8$ |
| Cloud water MVD (μm) | Default | 6.24 | 19.41 | 25.54 | 30.25 | 40.00 | 24.83 |
| | Climatology | 3.66 | 10.08 | 13.21 | 16.05 | 21.60 | 13.35 |
| | CAMS | 4.01 | 10.93 | 14.00 | 16.75 | 21.91 | 14.08 |
| Rain mass mixing ratio (g kg$^{-1}$) | Default | $1.45*10^{-5}$ | $3.82*10^{-4}$ | $1.36*10^{-3}$ | $3.31*10^{-3}$ | $1.74*10^{-2}$ | $2.59*10^{-3}$ |
| | Climatology | $1.22*10^{-5}$ | $1.45*10^{-4}$ | $5.75*10^{-4}$ | $2.44*10^{-3}$ | $2.46*10^{-2}$ | $2.29*10^{-3}$ |
| | CAMS | $1.23*10^{-5}$ | $1.48*10^{-4}$ | $5.18*10^{-4}$ | $2.27*10^{-3}$ | $2.36*10^{-2}$ | $2.12*10^{-3}$ |
| Cloud ice mass mixing ratio (g kg$^{-1}$) | Default | $1.27*10^{-7}$ | $6.99*10^{-7}$ | $2.41*10^{-6}$ | $5.57*10^{-6}$ | $8.01*10^{-5}$ | $6.30*10^{-6}$ |
| | Climatology | $1.80*10^{-7}$ | $1.90*10^{-6}$ | $4.93*10^{-6}$ | $2.07*10^{-5}$ | $8.43*10^{-5}$ | $1.30*10^{-5}$ |
| | CAMS | $1.79*10^{-7}$ | $1.91*10^{-6}$ | $4.91*10^{-6}$ | $2.06*10^{-5}$ | $8.25*10^{-5}$ | $1.27*10^{-5}$ |

The above speculation is further supported by analysing the source and sink terms of cloud water. Figure 5 shows the vertical profiles of cloud water mass tendency in the in-flight icing region between 12:00 and 14:00 UTC. The names of the source and sink terms related to cloud water mass tendency are listed in Table 3. This period was chosen because it corresponds to the gradual increase in rain in the Default experiment (Fig. 4a). As shown in Fig. 5a, the primary sink term





for SLW is the auto-conversion from cloud water to rain, which is driven mainly by collision-coalescence processes. Accretion between cloud water and rain is a secondary sink term. The largest contributor to cloud water mass tendency is the water vapor condensation term. As a result, despite the effects of the two sink terms, the cloud water mass tendency remains positive, leading to a gradual increase in SLW. In contrast, for the Climatology and CAMS experiments (Fig. 5b, Fig. 5c),

the smaller MVD results in less efficient collision-coalescence processes. Consequently, the auto-conversion from cloud water to rain and the accretion between cloud water and rain are negligible, resulting in a rapid increase in SLW.

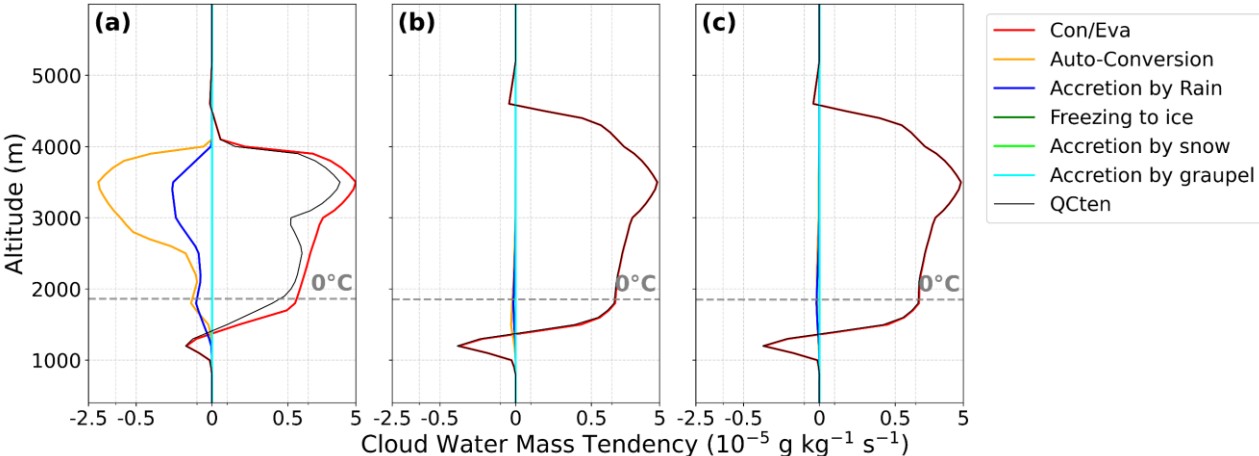

**Figure 5 Time-domain-average vertical profiles of cloud water mass tendency for Default (left), Climatology (middle) and CAMS (right) from 12:00 to 14:00 UTC.**

**Table 3 List of source and sink terms and their definitions related to cloud water mass tendency.**

| Name in microphysical scheme | Name in caption | Meaning |
|---|---|---|
| PRW_VCD | Con/Eva | Condensation and evaporation of cloud water |
| PRR_WAU | Auto-Conversion | Auto-conversion from cloud water to rain |
| PRR_RCW | Accretion by rain | Accretion between cloud water and rain |
| PRI_WFZ, PRI_HMF | Freezing to ice | Freezing of cloud water to cloud ice |
| PRS_SCW, PRG_SCW | Accretion by Snow | Accretion between cloud water and snow |
| PRG_GCW | Accretion by graupel | Accretion between cloud water and graupel |

In addition, the smaller size of SLW in the Climatology and CAMS experiments may also contribute to the pronounced increase in cloud ice in the upper levels and rain in the lower levels after 18:00 UTC, accompanied by a corresponding reduction in SLW (Fig. 4b-c). As noted by Pruppacher et al. (1998), small supercooled droplets are more efficiently collected

by ice crystals or snow, thereby accelerating the depletion of SLW. Moreover, their larger surface-area-to-volume ratio makes them more prone to evaporation, supplying more water vapor to the surrounding environment and serving as primary contributors to the Wegener-Bergeron-Findeisen process. This conclusion is further supported by the source and sink terms of cloud water shown in Fig. 6. Compared to the Default experiment, the Accretion between cloud water and snow term is



significantly larger in the Climatology and CAMS experiments (Fig. 6b-c). Together with the cloud water evaporation term,
these processes lead to the depletion of SLW below 3500 m.

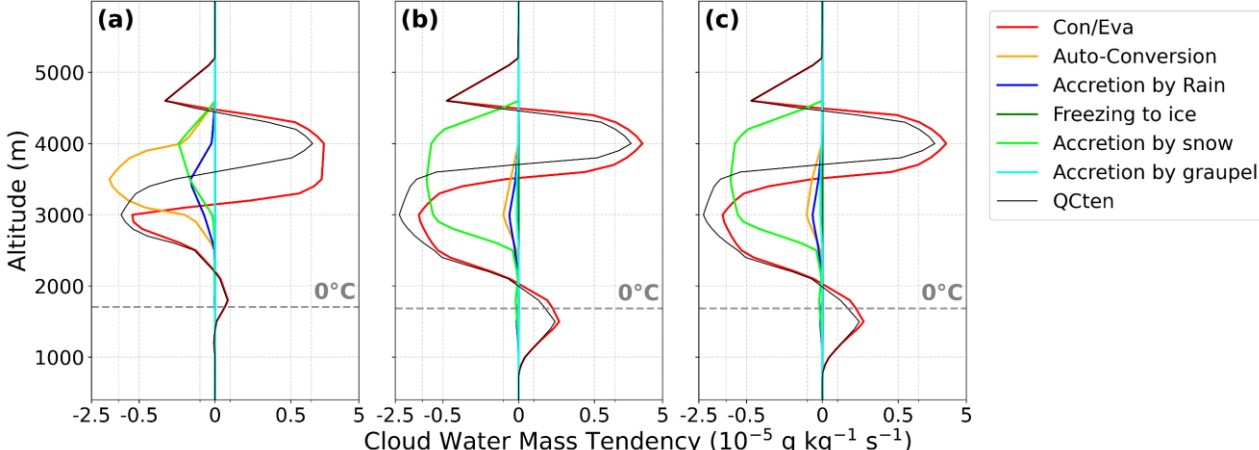

**Figure 6 Same as Fig.5 but for 21:00 to 24:00 UTC.**

Figure 7 quantifies the distribution of SLW at different temperatures. In the Default experiment, the majority (99th percentile)
of SLW values are below $0.8\,\mathrm{g\,kg^{-1}}$, and the 50th percentile of simulated SLW remains around $0.15\,\mathrm{g\,kg^{-1}}$ regardless of
temperature. When the temperature is below $0\,°C$, the 99th percentile of SLW decreases  markedly as the temperature
decreases, and only a small proportion of SLW is present at temperatures below $-10\,°C$. This phenomenon was also reported
by Thompson et al. (2017, Fig. 2), who explained that once ice forms at lower temperatures, cloud glaciation accelerates,
thereby reducing the amount of SLW.

When the temperature is above $0\,°C$, the cloud water mass mixing ratio increases rapidly with decreasing temperature and
reaches its maximum near the $0\,°C$ isotherm, as water vapor preferentially condenses in this thermal transition zone.
Although aerosol concentrations are typically highest near the surface (Fig. 2a), CCN can only be activated into cloud
droplets after being lifted to altitudes corresponding to the $0\,°C$ level, where they encounter supersaturated conditions.

In the Climatology and CAMS experiments (Fig. 7b-c), all percentiles of SLW are higher than those in the Default
experiment, with the 99th percentile of simulated SLW reaching approximately $0.9\,\mathrm{g\,kg^{-1}}$. Unlike the Default experiment,
the Climatology and CAMS experiments show that when the temperature is above $-7\,°C$, SLW values below the 75th
percentile increase with decreasing temperature, but decline sharply below $-7\,°C$. This suggests that above $-7\,°C$, the
enhancement of SLW due to the first aerosol indirect effect is sufficient to offset the depletion of SLW caused by cloud
glaciation. Additionally, when the temperature is around $-9\,°C$, SLW in the Climatology and CAMS experiments exhibits a
secondary peak. This is attributed to the first aerosol indirect effect not only increasing SLW but also deepening the cloud
system (see Fig. 4b-c), thereby producing a second SLW peak at a higher altitude corresponding to $-9\,°C$.





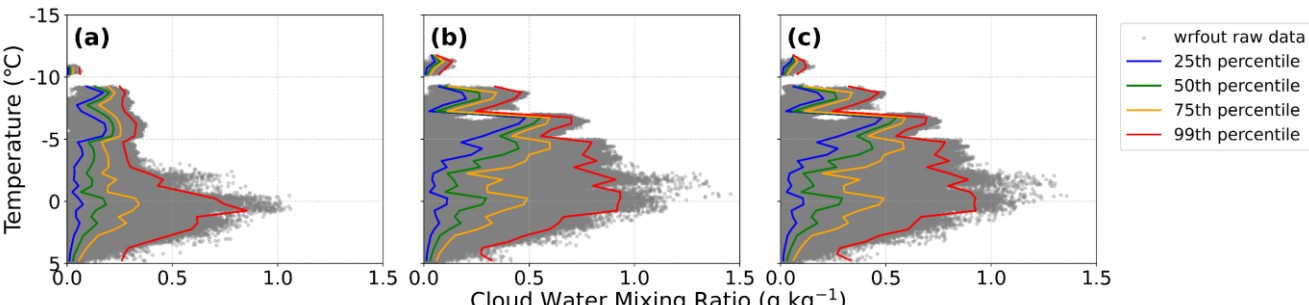

**Figure 7 Scatterplot of temperature (°C) and cloud water mass mixing ratio (g kg⁻¹) with the 25th, 50th, 75th, and 99th percentiles for each degree Celsius for the three experiments: Default (left), Climatology (middle) and CAMS (right).**

Figure 8 offers an alternative perspective on the distribution of SLW across different temperature ranges. All three experiments exhibit a rapid decrease in the relative frequency of SLW with every 5 °C drop in temperature. In the clean environment (Fig. 8a), SLW displays a narrow distribution, with the vast majority of values below 0.5 g kg⁻¹. In contrast, in polluted environments (Fig. 8b-c), the frequency of higher SLW values (greater than 0.5 g kg⁻¹) is greater than that in the clean environment, while the frequency around 0.25 g kg⁻¹ is notably lower.

Another notable feature under polluted conditions is the bimodal distribution of SLW within the temperature range of −5 °C to −10 °C. The peak at 0.25 g kg⁻¹ in the 0 °C to −5 °C range is accompanied by opposing valleys, with two distinct peaks, one at approximately 0.3 g kg⁻¹ and the other near 0 g kg⁻¹, indicating a relatively higher frequency of low SLW values. In the −10 °C to −5 °C range, SLW remains consistently low across all three experiments, suggesting that cloud development does not extend SLW to higher altitudes. Moreover, the relative frequency of cloud water in the 0 °C to 5 °C range peaks near 0 g kg⁻¹, indicating a scarcity of cloud water at these temperatures. This frequency is generally lower than that observed in the 0 °C to −5 °C range, further supporting the conclusion in Fig. 7, that CCN must be fully lifted to altitudes near the 0 °C level and reach supersaturation to be activated into cloud droplets and subsequently enhance SLW.

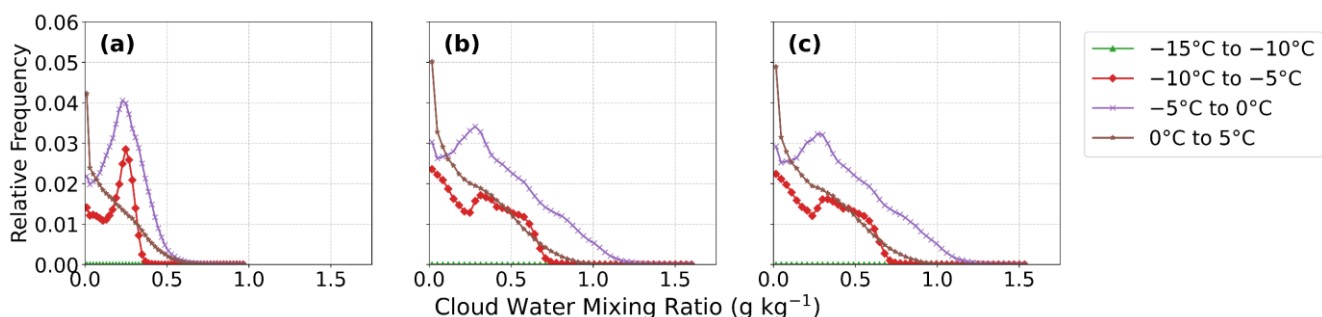

**Figure 8 Relative frequency of occurrence of specific ranges of cloud water mass mixing ratio (g kg⁻¹) in 5°C intervals of temperature for the three experiments: Default (left), Climatology (middle) and CAMS (right).**



### 3.3 Comparative assessment of simulated and observed SLW

We first focus on the intriguing phenomena highlighted by the violin and box plots of aerosol and cloud MVD based on PCASP and CDP data (Fig. 9). Below 3000 m, the distributions of the MVD for both aerosols and cloud droplets exhibit a unimodal pattern, with aerosol MVD peaking around 0.3 µm and cloud droplet MVD peaking around 4 µm. This indicates that cloud droplets are primarily in the initial stage of activation and condensational growth (Twomey, 1959). During this stage, the abundant aerosol particles in the lower layers, particularly those around 0.3 µm, competes for limited water vapor, resulting in slower droplet growth, a relatively uniform droplet size spectrum, and cloud droplet sizes primarily concentrated blow 5 µm. Above 3000 m, the MVD distributions of both aerosols and cloud droplets exhibit a clear bimodal pattern. The first peak corresponds to the peak observed below 3000 m. The second peak appears at approximately 2 µm for aerosols and around 15-18 µm for cloud droplets, indicating the presence of larger droplets. This suggests that as the cloud develops and droplets ascend, they continue to grow through collision-coalescence processes, leading to a broadening of the droplet size spectrum (Beard and Ochs, 1993; Seifert and Beheng, 2006; Hoffmann and Feingold, 2023). Additionally, the larger cloud droplets at the second peak may also be formed by the activation of CCN from aerosols at the second peak (Pruppacher et al., 1998).

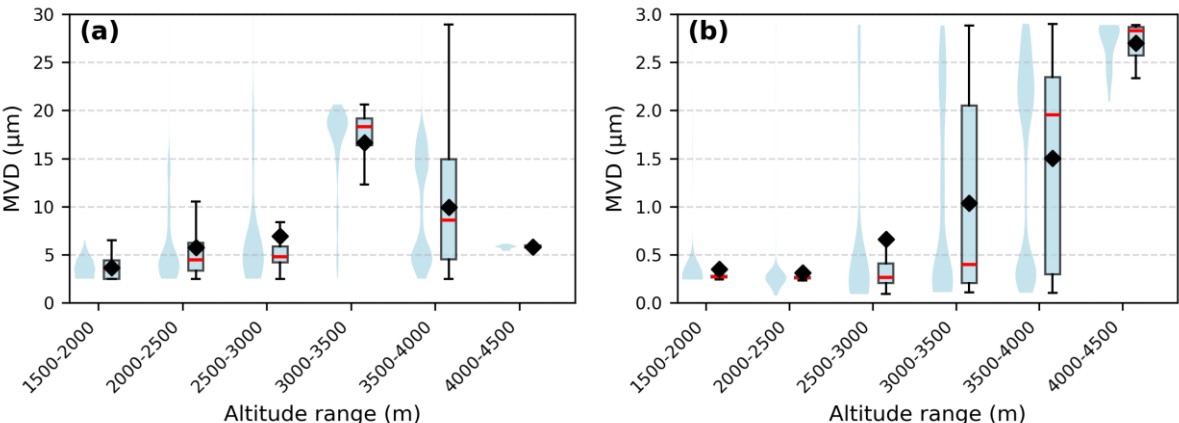

**Figure 9 Altitude-resolved violin and box plots of MVD for cloud droplets (left) and aerosols (right), in 500 m bins, derived from CDP and PCASP data (The diamonds indicate the mean values, and the red lines denote the medians).**

To evaluate the accuracy of simulated SLW properties across different numerical experiments, the simulated SLW within the flight icing region is compared with CDP observations. A time window from 12:00 to 12:30 and an altitude range of 3500-3700 m are selected to match observational and model conditions. Figure 10 presents scatter plots of SLW content and MVD derived from CDP measurements (3257 data points) and the three numerical experiments. The CDP observations (black dots) clearly confirm the bimodal MVD distribution identified in Fig. 9a, with distinct peaks at approximately 4 µm and 15-18 µm. Relative to the CDP data, the scatter from the Default experiment is evidently shifted to the right (Fig. 10a), with a maximum MVD approaching 40 µm, far exceeding the second observed peak, indicating that the Default experiment significantly overestimates MVD. In contrast, the Climatology experiment produces MVD values slightly left of the CDP's second peak





(Fig. 10b), implying an underestimation due to excessive aerosol concentrations that suppress droplet growth. The CAMS experiment, while slightly overestimating SLW content, aligns closely with the second MVD peak (Fig. 10c), suggesting improved representation of cloud droplet size. However, all three experiments fail to reproduce the first MVD peak near 4 µm, highlighting a common limitation in representing small droplet populations.

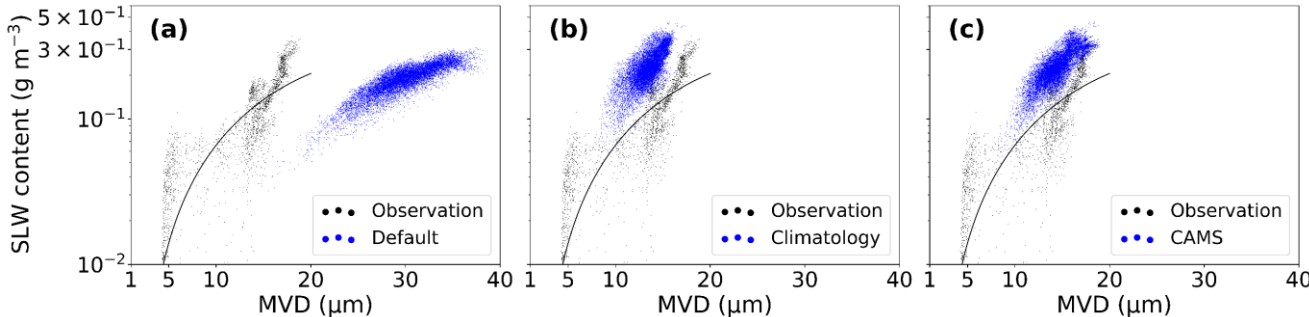

**Figure 10 Scatter plots of observed (black points) and simulated (blue points) SLW content versus MVD for the three experiments: Default (left), Climatology (middle), and CAMS (right). The black line represents the cubic curve fitted to the observed data points.**

**Table 4 The percentiles of the content, number concentration, and MVD of SLW for the Default, Climatology, CAMS experiments, and CDP observation.**

| SLW | Experiments and observation | Percentiles | | | | | |
|---|---|---|---|---|---|---|---|
| | | 1th | 25th | 50th | 75th | 99th | Mean |
| Content (g m$^{-3}$) | Default | 0.040 | 0.099 | 0.123 | 0.153 | 0.190 | 0.123 |
| | Climatology | 0.063 | 0.117 | 0.147 | 0.175 | 0.215 | 0.145 |
| | CAMS | 0.066 | 0.116 | 0.145 | 0.173 | 0.216 | 0.144 |
| | CDP | 0.011 | 0.057 | 0.103 | 0.167 | 0.320 | 0.120 |
| Number concentration (cm$^{-3}$) | Default | 8.03 | 11.14 | 13.23 | 15.11 | 19.58 | 13.28 |
| | Climatology | 90.11 | 136.58 | 160.30 | 191.98 | 340.64 | 172.11 |
| | CAMS | 87.74 | 121.08 | 139.73 | 162.95 | 236.67 | 144.37 |
| | CDP | 0.08 | 12.27 | 460.11 | 608.37 | 1254.11 | 401.92 |
| MVD (µm) | Default | 16.70 | 24.77 | 27.03 | 29.69 | 34.53 | 27.09 |
| | Climatology | 9.34 | 11.41 | 12.74 | 13.89 | 16.20 | 12.65 |
| | CAMS | 9.61 | 12.18 | 13.28 | 14.46 | 16.24 | 13.24 |
| | CDP | 2.50 | 4.61 | 12.24 | 15.16 | 20.50 | 10.51 |

The comparison of percentiles of SLW characteristics simulated by the three experiments further emphasizes the differences in the simulation of SLW properties (Table 4). Although the mean and median SLW values simulated by the Default experiment are closer to the observations (with the CDP mean at 0.12 g m$^{-3}$ and corresponding means of 0.123, 0.145, and





0.144 g m$^{-3}$ in the Default, Climatology, and CAMS experiments, respectively), this apparent agreement is primarily a result of compensating errors commonly seen in numerical simulations (Paukert et al., 2019; Zhao et al., 2022). Specifically, while
the Default experiment reproduces a seemingly accurate SLW content, it significantly overestimates the MVD of cloud droplets (27.09 μm compared to the observed 10.51 μm). This overestimation compensates for the severe underestimation of number concentration (13.28 cm$^{-3}$ compared to the observed 401.92 cm$^{-3}$), thereby masking the deficiencies in the simulation of microphysical properties. Additionally, the Default experiment is less effective than the Climatology and CAMS experiments in simulating the SLW peak. Specifically, the 75th and 99th percentile of SLW in the Climatology and
CAMS experiments is closer to the observed value, indicating that these experiments capture the higher extremes of SLW content more effectively.

In terms of number concentration, the simulated values in the polluted environment are significantly greater than those in the clean environment. Although the initial aerosol concentrations in the lower layers of the Climatology and CAMS experiments are higher than the PCASP observations (Fig. 2), the simulated SLW number concentrations above the 25th
percentile are much lower than the observed values. This discrepancy is a common issue in numerical models and is referred to as the CDNC bias (Sotiropoulou et al., 2006). This issue is further explored in the Sect. 4.

In terms of MVD, the simulated values in the polluted environment are closer to the observed ones, with the 25th and 99th percentiles at 11.4 and 16.2 μm, respectively. This aligns with the findings from the FAA icing database and Sand et al. (1984), which report that 75% of MVD values fall within the range of 10-20 μm. The median and mean values simulated by
the Climatology experiment are more consistent with the observations, while the 75th and 99th percentiles simulated by the CAMS experiment are closer to the observed values. This suggests that CAMS provides a better simulation for larger droplets, as indicated by the blue dot in Fig. 10c, which coincides with the observed second peak of MVD. However, all experiments fail to simulate the peak of MVD at 5 μm. This is due to the Thompson-Eidhammer aerosol-aware microphysics scheme, a bulk microphysics scheme, which assumes that the droplet size distribution of cloud water follows a generalized
gamma distribution. This assumption cannot accurately capture the multi-peak droplet size distribution that may be observed in reality.

In a comprehensive comparison, although the CAMS experiment slightly overestimates the mean SLW content and exhibits a larger deviation in SLW number concentration compared to the Climatology experiment, it demonstrates superior capability in capturing high SLW values and larger MVDs. Elevated SLW content and the presence of large supercooled
droplets are critical factors contributing to severe in-flight icing, highlighting the importance of accurately representing these features in numerical simulations.

## 4 Conclusions

This study investigates the impact of ACI on the properties of SLW during in-flight icing events in the high-aerosol-concentration environment of the Sichuan Basin. Using the Thompson-Eidhammer aerosol-aware microphysics scheme,



three different initial aerosol number concentrations are designed. The numerical experiments show that all three experiments reproduce the synoptic-scale spatial distribution of SLW. Compared with simulations in a clean environment, those in a polluted environment simulate higher SLW mass mixing ratios, more number concentrations, smaller droplet sizes, and longer cloud system lifetimes. The experiments also reveal that in clean environments, a stronger auto-conversion process suppresses SLW formation, whereas in polluted environments, enhanced accretion process accelerates SLW

depletion. Furthermore, the experiments demonstrate that the enhanced first aerosol indirect effect in polluted conditions can partially offset the depletion of SLW caused by glaciation. Observational data further reveal distinct cloud droplet growth mechanisms at different altitudes, with condensational growth dominating in the lower layers and collision–coalescence growth prevailing in the upper layers.

The study highlights the widespread occurrence of CDNC bias between numerical experiments and observations (Hoose et

al., 2009; Betancourt and Nenus, 2014). One key reason for this bias could be the limitations of the CCN activation parameters in the model (Weston et al., 2022). In the Thompson-Eidhammer aerosol-aware microphysics scheme, CCN activation is based on a lookup table or a parameterized formula that considers factors such as vertical velocity, activation threshold, and aerosol particle size. If the model underestimates the updraft at the cloud base or within the cloud, it will not activate enough cloud droplets, even when there are sufficient CCN available. Furthermore, an overestimated auto-

conversion processes in the model, resulting in excessive depletion of supercooled droplets, together with scale mismatches between observations and model resolution, can also contribute to CDNC bias. These findings suggest that future research should explore these issues from multiple angles to improve the representation of CDNC in numerical models.

Another noteworthy phenomenon is that, despite the increase in the number concentration of IN in the Climatology experiment and the decrease in the CMAS experiment, neither experiment simulates ice-phase particles during the SLW

increase phase (12:00-20:00 UTC), which contradicts the observations. In our previous study (Yuan et al., 2025, Fig. 7), CIP imagery shows the presence of ice-phase particles during this stage, with the ice water content at an altitude of 3500 m being on the order of $10^{-3}$ g m$^{-3}$ (not shown). This discrepancy suggests that the Thompson-Eidhammer aerosol-aware microphysics scheme does not accurately describe the parameterization of IN, as also confirmed by the study of Wu et al. (2024). Future studies will aim to improve the parameterization of IN or evaluate its parameterization in other microphysical

schemes. Additionally, the polluted environment produces more ice and snow than the clean environment during the SLW reduction stage (after 20:00 UTC). This may be due to the decrease in temperature, as the altitude of the −10°C isotherm in Fig. 4 drops significantly after 18:00 UTC. However, there is a lack of observational verification for this stage.

Due to the limitations of the bulk microphysics scheme, none of the three numerical experiments accurately reproduce the observed bimodal droplet size distribution. While the CAMS experiment demonstrates a better ability to capture high SLW

values and larger MVD, it fails to represent cases with low SLW content and small droplet sizes. In future studies, bin microphysics schemes will be employed to better capture the characteristics of the bimodal droplet size distribution of SLW.

Finally, it should be noted that although this study utilizes near-real-time aerosol data from CAMS, the method used to calculate aerosol number concentration still carries considerable uncertainties and lacks validation against observational

evidence. Accurately characterizing aerosol emission inventories and quantifying their spatiotemporal distribution in the
atmosphere remain significant challenges. These issues are critical for advancing our understanding of ACI and improving
the representation of SLW in numerical models.

## Data availability

The CAMS reanalysis datasets are obtained from https://ads.atmosphere.copernicus.eu/datasets/cams-global-atmospheric-composition-forecasts?tab=download, the three-dimensional monthly averaged aerosol climatology file provided by WPS
are obtained from https://www2.mmm.ucar.edu/wrf/users/physics/mp28_updated.html.

## Author contribution

MY contributed to conceptualization, formal analysis, funding acquisition, creation of models, computing resources,
programming, and writing. DW contributed to formal analysis, programming, visualization, and original draft writing. WW
and DZ contributed to aircraft measurements data collection. LY contributed to creation of models and designing computer
programmes. XD contributed to data curation. FP contributed to critical review of writing .

## Competing interests

The authors declare that they have no conflict of interest.

## Acknowledgments

This work is supported by the Fundamental Research Funds for the Central Universities (grant no. 24CAFUC01003),
National Natural Science Foundation (grant no. U1333130), China Meteorological Administration (CMA) Innovation
Development Special Program (grant no. CXFZ2025J038), Capacity Building for Weather Modification in Southwest
China-Research and Experiment Project on  Detection of Stratocumulus-Cumulus Mixed Clouds and Convective Clouds in
Complex Terrain and Artificial Catalysis Technology (grant no. SCIT-ZG(Z)-2024100001-3). The in situ aircraft
measurements data used in this study were obtained from China Meteorological Administration Weather Modification
Centre and Weather Modification Office of Sichuan Province.

## Financial support

This research has been supported by the Fundamental Research Funds for the Central Universities (Grant Number.
24CAFUC01003)



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
