# Peer review of "Enhanced Simulation of Supercooled Liquid Water for In-Flight Icing Using an Aerosol-Aware Microphysics Scheme with CAMS Reanalysis"

_EGUsphere, 2025_

## Referee Comment (RC2)

**Review of: "Enhanced Simulation of Supercooled Liquid Water for In-Flight Icing Using an Aerosol-Aware Microphysics Scheme with CAMS Reanalysis" by Yuan et al.**

**Overall impression and rating**

The manuscript describes three model simulations of supercooled liquid clouds with different aerosol schemes as input. The results are compared with each other and with aircraft observations. Overall, the manuscript is very easy to understand, well structured, and clearly written. Although the actual results do not necessarily provide novel scientific findings, they nevertheless show a robust comparison of three aerosol schemes and their effects on the results with regard to supercooled liquid clouds. This comparison may be particularly useful for the modeling community in terms of further optimizing models. I therefore recommend the manuscript for publication in ACP after my more technical comments have been taken into account.

**Specific comments/questions:**

- Title: I suggest changing the title of the manuscript so that it better reflects the actual comparisons of a model with three different aerosol input datasets.

- line 39: I would recommend pointing out the major differences between INPs and CCNs here. In particular, INPs are always solid, insoluble particles, while CCNs are always water-containing liquid aerosols (e.g., Belosi et al. 2017 or Krämer et al. 2016). Therefore, I would recommend writing the following in line 39: ...insoluble aerosol act as...

- Section 2.5: How were the in situ measurements compared exactly with the model output? Section 2.2 states that only out-cloud sections of the measurements were used. Was this (only out cloud) also done with the model results? Otherwise, the concentrations (inside and outside clouds) are not necessarily comparable. This is particularly important with regard to the section in lines 387-391 and should also be taken into account in the discussion.

- Section: 3.1: Intercomparison to ERA5 data. First, the ERA5 data should be introduced in Section 2 because you used it in the end to compare the other simulations to ERA5. And you can't assume that everyone knows the details of ERA5 relevant for this comparison.

- Line 368: I would perhaps simply add that the models lack in reproducing a bimodal SLW distribution.

**Technical comments/questions:**

- line 62: "simply" to "simple"

- line 87: "climatic" to "climatological"

- line 108: "is" to "were"

- Figure 1: The arrow of the flight path is hardly visible. Maybe place it better directly next to flight path or enlarge it a bit. In addition, it would be nice, if you could also plot the innermost nest in panel b. Then it is easier to compare.

- Figure 2: The orange line looks more reddish. May name it red instead of orange.

- Figure 4: I would recommend collecting the explanation of the isolines in one place in the caption. Explain the part: cloud water: g kg-1, rain: 10-3 g kg-1, cloud ice: 10-7 g kg-1, snow: 10-3 g kg-1, temperature: °C together with the shading.

- Table 2: Please avoid line breaks in the units (row 1 and 5).

- Figure 5: Please provide information in the caption, what QCten is. All other classes can be inferred by the name itself.

- line 361: Please exchange "confirm" with "show" or similar. Because you cannot confirm the same thing with the same data. Both plots show this feature in MVD from the same CDP data.

- Line 397: Which blue dot are you referring to in Figure 10c. I just see only multiple blue plots from simulation, but not a specific one.

**References:**

- Belosi, F., Rinaldi, M., DeCesari, S., Tarozzi, A., Nicosia, A. and Santachiara, G. (2017) Ground Level Ice Nuclei Particle Measurements Including Saharan Dust Events at a Po Valley Rural Site (San Pietro Capofiume, Italy). Atmospheric Research, 186, 116-126. https://doi.org/10.1016/j.atmosres.2016.11.012

- Krämer, M., Rolf, C., Luebke, A., Afchine, A., Spelten, N., Costa, A., Meyer, J., Zöger, M., Smith, J., Herman, R. L., Buchholz, B., Ebert, V., Baumgardner, D., Borrmann, S., Klingebiel, M., and Avallone, L.: A microphysics guide to cirrus clouds – Part 1: Cirrus types, Atmos. Chem. Phys., 16, 3463–3483, https://doi.org/10.5194/acp-16-3463-2016, 2016.

---

## Author Comment (AC1)

This work compares the performance of different WRF simulation configurations on the accurate depiction of supercooled liquid water by comparing a particular event to flight observations. Overall, the manuscript is well written, it presents the methods and results clearly, and gives a good and thorough description of the model limitations that may explain the biases in the resulting performance. It seems like a novel simulation configuration using CAMS data is performed, thus its benchmarking is the main contribution of the study. The manuscript could benefit from a better explanation of its novelty and proposing future work to overcome the reported biases of the new approach.

We sincerely appreciate the reviewers' valuable comments, which have greatly strengthened the manuscript. We have revised the manuscript accordingly and believe that the revised version is now more complete. The rewritten abstract highlights the key innovations of this study, and the conclusion outlines potential improvements to address the biases associated with the new approach in future work. Please refer to our point-by-point responses in blue text, and note that revisions in the manuscript are indicated in italic text.

Minor comments

-Please revise the title, as the core of the manuscript does not reflect its focus

We have revised the title to "*Quantitative Assessment of Supercooled Liquid Water Sensitivity to Different Aerosol Field Inputs over the Sichuan Basin*", which we believe effectively captures the focus of the study.

- The abstract does not clearly explain which simulation configuration is novel, and it also does not clearly explain which are clean and polluted conditions. Also, it is not clear if the synoptic scale is compared with a reference

We consider it a novelty of this study to quantitatively assess the properties of supercooled liquid water using different aerosol inputs and to represent near-real-time aerosol conditions with typical aerosol number densities and size distributions characteristic of East Asia. We also provide clear definitions of clean and polluted conditions, and explicitly use ERA5 as a reference for the synoptic-scale distribution of supercooled liquid water. The revised abstract is as follows:

*Abstract. Aerosol-cloud interactions profoundly influence the formation and evolution of supercooled liquid water, a key factor in in-flight icing. However, accurately quantifying aerosol emission inventories and their spatiotemporal distributions remains a major challenge, particularly in high-aerosol environments such as the Sichuan Basin in China. In this study, the Thompson–Eidhammer aerosol-aware microphysics scheme is applied to a high-aerosol icing event to quantitatively assess the sensitivity of supercooled liquid water properties to different aerosol inputs. Three aerosol configurations are examined: (1) the scheme's default settings representing clean conditions; (2) climatological aerosol values representing polluted conditions; and (3) near-real-time aerosol fields from the Copernicus Atmosphere Monitoring Service (CAMS), which are converted from mass to number concentrations using typical densities and size parameters for major East Asian aerosol species, representing polluted and more realistic environments. All simulations reproduce the synoptic-scale supercooled liquid water and temperature distribution when compared with ERA5. Relative to clean conditions, polluted-environment simulations produce higher supercooled liquid water content, larger cloud droplet number concentrations, smaller median volume diameters, and longer cloud lifetimes. The experiments also reveal that stronger auto-conversion in clean conditions suppresses supercooled liquid water formation, whereas enhanced riming in polluted environments promotes supercooled liquid water depletion. In situ aircraft observations further indicate that the CAMS-driven experiment performs best in capturing the high supercooled liquid water content and large median volume diameters. These findings underscore the importance of near-real-time aerosol inputs for improving simulations of aerosol-cloud interactions and predicting aircraft-icing environments.*

- The research gap (L79-L86) is presented in terms of a previous study of these authors. This should be improved by informing the state of the art of all relevant works that have similar research initiatives. Similarly, while the topic of interest is mentioned, the simulation configurations are not justified

In the revised manuscript, we have emphasized the importance of accurately representing aerosol particle size and number concentration distributions, expanded the discussion of existing research gaps, and summarized the current progress in this field. In addition, we provide further justification for the

simulation configurations, explaining the rationale for selecting the different experimental setups and citing relevant references:

*In recent years, numerical models have integrated several double-moment microphysics schemes to simulate ACI, yielding promising advancements in the representation of cloud microphysics (Glotfelty et al., 2019). These schemes typically rely on fixed, prescribed, or simply parameterized aerosol inputs, such as aerosol number concentration, size distribution, and chemical composition, to estimate the number of particles activated as CCN, which subsequently form SLW. Because these schemes do not require complex emission inventories or detailed chemical boundary conditions, they are well suited for proof-of-concept simulations and sensitivity experiments investigating the effects of aerosol perturbations on cloud microphysics. A representative example is the Thompson–Eidhammer aerosol-aware scheme. Thompson and Eidhammer (2014) first implemented this scheme and assessed precipitation in a large winter cyclone using empirical climatological aerosol emission data. The scheme is then applied to simulate in-flight icing, assessing key characteristics such as liquid water content, median volume diameter (MVD), cloud droplet number concentrations (CDNC), and temperature in aircraft icing environments (Thompson et al., 2017). Weston et al. (2022) apply the scheme to evaluate the influence of different initial CCN number concentrations during two fog events in Namibia. Thomas et al. (2021) use the scheme to successfully capture the spatial and temporal evolution of an extreme rainfall event in southwestern India. Subsequent studies aim to enhance simulation performance by incorporating more realistic aerosol emissions.*

*Previous studies have shown that the chemical composition of aerosols has a much smaller influence on ACI than their concentration and particle size (Dusek et al., 2006; Ward et al., 2010). Therefore, continuously improving the representation of aerosol particle size and number concentration distributions in models is crucial. Li et al. (2024) incorporated aerosol fields from the Common Community Physics Package to improve precipitation simulations over Europe and North America. He et al. (2025) converted CAMS aerosol mass mixing ratios into particle number concentrations under different aerosol concentration background using a lognormal distribution assumption, and further investigated how variations in microphysical processes within the Southwest Vortex over China influence precipitation patterns. Wu et al. (2024) improved the simulation of SLW by*

*incorporating aerosol concentrations for different particle size bins derived from the Community Multiscale Air Quality model.*

*Several studies have examined the role of aerosols in SLW formation and icing processes, yet few have explicitly evaluated aerosol-aware microphysics schemes under uniquely high-aerosol conditions. Building on our previous analysis of in-flight icing events over the Sichuan Basin (Yuan et al., 2025), which revealed higher aerosol concentrations, greater CDNC, and smaller cloud droplet sizes compared with typical icing environments, this study aims to systematically assess the performance of an aerosol-aware microphysics scheme in simulating SLW properties under such conditions, thereby addressing this research gap. We employ the Thompson–Eidhammer aerosol-aware microphysics scheme and design three aerosol input configurations to evaluate its performance and sensitivity: (1) the scheme's default aerosol settings as a baseline; (2) climatological aerosol values representing typical background conditions; and (3) near-real-time aerosol data from the Copernicus Atmosphere Monitoring Service (CAMS). This configuration strategy allows the experiments to span a realistic range of aerosol environments and enables a quantitative assessment of how aerosol variability influences SLW production. In particular, the use of CAMS data provides physically consistent aerosol loading fields, thereby enhancing the realism of the simulations and enabling diagnosis of limitations in the default parameterization under high-aerosol conditions.*

**References**

Dusek, U., Frank, G. P., Hildebrandt, L., Curtius, J., Schneider, J., Walter, S., ... & Andreae, M. O.: Size matters more than chemistry for cloud-nucleating ability of aerosol particles, Science, 312(5778), 1375-1378, 2006.

Ward, D. S., Eidhammer, T., Cotton, W. R., and Kreidenweis, S. M.: The role of the particle size distribution in assessing aerosol composition effects on simulated droplet activation, Atmos. Chem. Phys., 10, 5435–5447, https://doi.org/10.5194/acp-10-5435-2010, 2010.

He, Y., Zhao, P., Xiao, H., & Zhao, C.: Structural difference on the response of microphysical and precipitation processes to aerosol perturbation in a quasi-stationary Southwest Vortex system, J. Geophys. Res. Atmos., 130(1), e2024JD041767, https://doi.org/10.1029/2024JD041767, 2025.

- If I understand correctly, only the initial conditions are varied within all the experiments. Does this mean that there are no emissions during the simulation? Could this lead to specific biases when comparing to the observations?

In the Thompson–Eidhammer aerosol-aware microphysics scheme, a constant-in-time fake surface aerosol emission (or flux/tendency) is calculated as a two-dimensional field using the mean surface wind and the initial near-surface number concentration of water-friendly aerosols (WFA), according to the following relation applied only at the model's lowest level:

$$dN_{wfa}/dt = 10^{[\log(N_{wfa})-3.69897]}$$

In contrast, no surface emission tendency is applied for ice-friendly aerosol (IFA) (Thompson and Eidhammer, 2014).

- When presenting the results, ERA5 reanalysis data performs poorly. What could be causing such poor performance in a reanalysis product?

We found that, compared with ERA5, the numerical simulation performs reasonably well in capturing the synoptic-scale temperature field and the distribution of supercooled liquid water (SLW). However, the locations of the SLW maxima differ: the simulations place the peak SLW west of the icing region, along the mountainous margins of the basin, whereas ERA5 does not exhibit these maxima.

This discrepancy primarily arises from two factors. First, the coarse spatial resolution of ERA5 limits its ability to resolve terrain-induced local ascent, condensation, and mixed-phase microphysical processes. Orographic lifting generates pronounced windward–leeward contrasts and strong localized circulations that govern where liquid water forms; however, these features are overly smoothed or displaced in ERA5. As a result, representativeness errors occur, leading to biases in the liquid water content. Moreover, the simplified cloud and microphysical parameterizations used in global reanalyses further exacerbate these biases in mountainous regions.

Second, the ERA5 dataset is produced by assimilating diverse observational data into a numerical weather prediction model, and the accuracy of cloud water depiction depends heavily on the

assimilated observations. In regions with complex terrain, the scarcity of observational data poses significant challenges for the assimilation process, ultimately reducing the reliability of the retrieved cloud water fields.

We have revised the second paragraph of Sect. 3.1 as follows:

*"The synoptic-scale distributions of temperature and SLW path in ERA5 (Fig. 3d) are generally consistent with those in the three numerical experiments, but the locations of the SLW maxima differ. The simulations place the peak SLW west of the in-flight icing region along the mountainous margins of the basin, whereas ERA5 does not capture these maxima. This discrepancy is primarily attributable to the relatively coarse spatial resolution of ERA5 (approximately 30 km), which is insufficient to resolve the physical processes and atmospheric dynamics associated with topographically induced local ascent over complex terrain. As a result, orographic uplift and the accompanying cloud structures are smoothed or displaced, leading to biases in the SLW path. In addition, the scarcity of observational data in mountainous regions poses challenges for the assimilation process, further reducing the reliability of the cloud water fields (Jiao et al., 2021; Wang et al., 2025; Hellmuth et al., 2025)."*

**References:**

Wang, J., Li, Z., Liang, Y., & Ke, J.: An Assessment of the Applicability of ERA5 Reanalysis Boundary Layer Data Against Remote Sensing Observations in Mountainous Central China, Atmosphere, 16(10), 1152, https://doi.org/10.3390/atmos16101152, 2025.

Jiao, D., Xu, N., Yang, F., & Xu, K.: Evaluation of spatial-temporal variation performance of ERA5 precipitation data in China, Sci. Rep., 11(1), 17956, https://doi.org/10.1038/s41598-021-97432-y, 2021.

Hellmuth, F., Carlsen, T., Daloz, A. S., David, R. O., Che, H., and Storelvmo, T.: Evaluation of biases in mid-to-high-latitude surface snowfall and cloud phase in ERA5 and CMIP6 using satellite observations, Atmos. Chem. Phys., 25, 1353–1383, https://doi.org/10.5194/acp-25-1353-2025, 2025.

- The last paragraphs in the discussion try to explain the differences between simulations and

observations, but the discussion is mainly descriptive. Based on the features that contribute to each difference, are there ideas that could improve future research? For instance, other microphysics schemes.

We mentioned future research plans in lines 427, 434, and 440, including the use of alternative microphysics schemes. In the revised manuscript, we have clarified this plan as follows:

*"Future research will focus on improving the parameterization of cloud droplet nucleation and ice nucleation, or evaluating the performance of other microphysics schemes."*

- Finally, these conclusions are all for a single case study. Can we consider this case study "normal" in order to generalize the conclusions? If not, how can you caution the readers about particular features that may not occur in other cases?

It should be acknowledged that the conclusions presented in this study are derived from a case-study framework. Regarding the generality of the finding that "polluted environments contain more SLW and exhibit stronger riming than clean environments," we offer the following discussion:

*"Notably, our three numerical experiments show an apparently monotonic SLW increase with CCN concentration. However, extensive research on convective clouds indicates that cloud water often responds non-monotonic to aerosols, with an optimal cloud development occurring at intermediate CCN levels (Dagan et al., 2017; Deng et al., 2024; Jeon et al., 2018). When CCN concentrations are below this optimal range, increasing CCN enhances condensation and cloud water, whereas excessive CCN leads to numerous small droplets, enlarged total droplet surface area, and strengthened evaporation, ultimately reducing cloud water. Similarly, although our simulations indicate stronger riming under polluted conditions, previous studies have shown that riming efficiency depends on a balance among hydrometeor size, cloud droplet concentration, and collision kernel (Cui et al., 2011). Riming may be suppressed under both extremely clean conditions, where droplet concentrations are insufficient, and extremely polluted conditions, where droplets are too small for effective collection (Barthlott et al., 2022; Cheng et al., 2010; Cui et al., 2011). Because this study only contrasts two aerosol states (clean and polluted), it cannot capture the full aerosol spectrum or the potential peak responses of riming that would emerge under intermediate aerosol conditions."*

Barthlott, C., Zarboo, A., Matsunobu, T., & Keil, C.: Importance of aerosols and shape of the cloud droplet size distribution for convective clouds and precipitation, Atmos. Chem. Phys., 22, 2153–2172, https://doi.org/10.5194/acp-22-2153-2022, 2022.

Cheng, C. T., Wang, W. C., & Chen, J. P.: Simulation of the effects of increasing cloud condensation nuclei on mixed-phase clouds and precipitation of a front system, Atmos. Res., 96, 461–476, https://doi.org/10.1016/j.atmosres.2010.02.005, 2010.

Cui, Z., Davies, S., Carslaw, K. S., & Blyth, A. M.: The response of precipitation to aerosol through riming and melting in deep convective clouds, Atmos. Chem. Phys., 11, 3495–3510, https://doi.org/10.5194/acp-11-3495-2011, 2011.

Dagan, G., Koren, I., Altaratz, O., & Heiblum, R. H.: Time-dependent, non-monotonic response of warm convective cloud fields to changes in aerosol loading, Atmos. Chem. Phys., 17, 7435–7444, https://doi.org/10.5194/acp-17-7435-2017, 2017.

Deng, X., Fu, S., & Xue, H.: The Non-Monotonic Response of Cumulus Congestus to the Concentration of Cloud Condensation Nuclei, Atmosphere, 15, 1225, https://doi.org/10.3390/atmos15101225, 2024.

Jeon, Y. L., Moon, S., Lee, H., Baik, J. J., & Lkhamjav, J.: Non-monotonic dependencies of cloud microphysics and precipitation on aerosol loading in deep convective clouds: A case study using the WRF model with bin microphysics. Atmosphere, 9(11), 434, https://doi.org/10.3390/atmos9110434, 2018.

Line-by-line comments/suggestions

L108 measurements "were"

Corrected

Fig. 1: The mentioned arrow is not clear

Changed

L176 This is a generic description; it'd be more useful to close the sentence explaining what will be used in this work

At the end of the paragraph, we clarify the approach used in this study:

*"In this study, we adopt fixed background values and reanalysis datasets inputs."*

L215 derived by whom?

derived by Equation (4)

Fig. 4: Domain-averaged means, Flight path averaged here, correct?

The domain-averaged region refers to an area centered at 29°N, 105°E, extending 50 km in all directions, which encompasses the orange path shown in Figure 1b.

Table 2: Do these values correspond to statistics of all data in space and time? Or at particular timesteps? Or at specific domains as the flight path?

These values represent statistics over all simulated times and spatial domains.

L277 Specify what the above speculation is

The above speculation refers to the influence of droplet size on the efficiency of the collision-coalescence process. We revised the sentence to:

*"The above speculation regarding the influence of droplet size on the efficiency of the collision-coalescence process is further supported by analysing the source and sink terms of cloud water."*

Fig. 5. What is QCten? Not explained in the manuscript.

QCten represents the cloud water mass tendency (i.e., the net rate of change of the total cloud water mixing ratio), which is the sum of all source and sink terms, including condensation/evaporation, auto-conversion, accretion, and freezing. We have added this information to Table 3.

---

## Author Comment (AC2)

**Overall impression and rating**

The manuscript describes three model simulations of supercooled liquid clouds with different aerosol schemes as input. The results are compared with each other and with aircraft observations. Overall, the manuscript is very easy to understand, well structured, and clearly written. Although the actual results do not necessarily provide novel scientific findings, they nevertheless show a robust comparison of three aerosol schemes and their effects on the results with regard to supercooled liquid clouds. This comparison may be particularly useful for the modeling community in terms of further optimizing models. I therefore recommend the manuscript for publication in ACP after my more technical comments have been taken into account.

We sincerely thank the reviewer for the valuable comments and suggestions, which have significantly improved the manuscript. We have revised the manuscript accordingly and believe that the revised version is now more complete. Please refer to our point-by-point responses in blue text; revisions in the manuscript are indicated in italic text.

**Specific comments/questions:**

• Title: I suggest changing the title of the manuscript so that it better reflects the actual comparisons of a model with three different aerosol input datasets.

Thank you for your suggestion. We have revised the title to "*Quantitative Assessment of Supercooled Liquid Water Sensitivity to Different Aerosol Field Inputs over the Sichuan Basin*", which we believe effectively captures the focus of the study.

• line 39: I would recommend pointing out the major differences between INPs and CCNs here. In particular, INPs are always solid, insoluble particles, while CCNs are always water-containing liquid aerosols (e.g., Belosi et al. 2017 or Krämer et al. 2016). Therefore, I would recommend writing the following in line 39: ...insoluble aerosol act as...

Thank you for your suggestion. We have rewritten it as follows:

*"On one hand, solid, insoluble aerosol particles act as ice nuclei (IN, Belosi et al. 2017)..., On the other hand, hygroscopic aerosols particles act as cloud condensation nuclei (CCN)…"*

Belosi, F., Rinaldi, M., DeCesari, S., Tarozzi, A., Nicosia, A. & Santachiara, G.: Ground Level Ice Nuclei Particle Measurements Including Saharan Dust Events at a Po Valley Rural Site (San Pietro Capofiume, Italy), Atmos. Res., 186, 116-126, https://doi.org/10.1016/j.atmosres.2016.11.012, 2017.

• Section 2.5: How were the in situ measurements compared exactly with the model output? Section 2.2 states that only out-cloud sections of the measurements were used. Was this (only out cloud) also done with the model results? Otherwise, the concentrations (inside and outside clouds) are not necessarily comparable. This is particularly important with regard to the section in lines 387-391 and should also be taken into account in the discussion.

Thanks for the insightful comments. We note that out-of-cloud aerosol data are used only to construct a realistic background aerosol concentration field, whereas the CDP measurements used for model evaluation are obtained under in-cloud conditions. The reviewer's comment highlighted that the model data previously used for comparison (12:00–12:30 UTC, 3500–3700 m, and a 50-km radius centered at 29°N, 105°E, encompassing the orange flight path in Fig. 1b) may have included out-of-cloud grid points.

In the revised analysis, we therefore applied in-cloud filtering criteria consistent with the aircraft in situ observations (CDNC > 10 cm$^{-3}$ and LWC > 10$^{-3}$ g m$^{-3}$, Zhang et al., 2011) and excluded simulated data that did not satisfy these thresholds. This ensures that the model–observation comparison is restricted to in-cloud conditions. The corresponding text in Sect. 3.3 has been revised as follows:

*"In addition, in-cloud criteria consistent with the aircraft in situ measurements (CDNC > 10 cm$^{-3}$ and LWC > 10$^{-3}$ g m$^{-3}$; Zhang et al., 2011) are applied, and simulated data not satisfying these thresholds are excluded. This procedure ensures that the model-observation comparison is restricted to in-cloud conditions only."*

The updated statistics (values that differ from the previous version are highlighted in red) indicate that only the Default experiment shows minor changes, which do not alter the conclusions of this study. The revised results are summarized in the table below.

| SLW | Experiments or observation | Percentiles | | | | | |
| --- | --- | --- | --- | --- | --- | --- | --- |
| | | 1th | 25th | 50th | 75th | 99th | Mean |
| Content (g m⁻³) | Default | 0.040 | 0.098 | 0.121 | 0.149 | 0.190 | 0.121 |
| | Climatology | 0.063 | 0.117 | 0.147 | 0.175 | 0.215 | 0.145 |
| | CAMS | 0.066 | 0.116 | 0.145 | 0.173 | 0.216 | 0.144 |
| | CDP | 0.011 | 0.057 | 0.103 | 0.167 | 0.320 | 0.120 |
| Number concentration (cm⁻³) | Default | 10.12 | 11.60 | 13.53 | 15.26 | 19.64 | 13.68 |
| | Climatology | 90.11 | 136.58 | 160.30 | 191.98 | 340.64 | 172.11 |
| | CAMS | 87.74 | 121.09 | 139.73 | 162.95 | 236.67 | 144.37 |
| | CDP | 0.08 | 12.27 | 460.11 | 608.37 | 1254.11 | 401.92 |
| MVD (μm) | Default | 16.58 | 24.52 | 26.71 | 29.11 | 33.03 | 26.59 |
| | Climatology | 9.34 | 11.41 | 12.74 | 13.89 | 16.20 | 12.65 |
| | CAMS | 9.61 | 12.18 | 13.28 | 14.46 | 16.24 | 13.24 |
| | CDP | 2.50 | 4.61 | 12.24 | 15.16 | 20.50 | 10.51 |

Zhang, Q., Quan, J., Tie, X., Huang, M., & Ma, X.: Impact of aerosol particles on cloud formation: Aircraft measurements in China, Atmos. Environ., 45(3), 665-672, https://doi.org/10.1016/j.atmosenv.2010.10.025, 2011.

• Section: 3.1: Intercomparison to ERA5 data. First, the ERA5 data should be introduced in Section 2 because you used it in the end to compare the other simulations to ERA5. And you can't assume that everyone knows the details of ERA5 relevant for this comparison.

We have added Sect. 2.4 to provide a description of the ERA5 dataset used in this study:

*2.4 ERA5 data*

*In this study, European Centre for Medium-Range Weather Forecasts Reanalysis 5 (ERA5) data from the European Centre for Medium-Range Weather Forecasts (ECMWF) are employed as an independent reference to evaluate the WRF simulations. ERA5 is produced with a coupled data assimilation and forecasting system that combines multiple observational datasets with a global numerical weather prediction model, and can therefore be regarded as a reanalysis that closely*

*approximates the observed atmospheric state. ERA5 provides globally complete atmospheric fields with hourly temporal resolution and a horizontal grid spacing of 0.25° (Hersbach et al., 2020). The ERA5 variables used in this work include three-dimensional temperature, cloud liquid water content, and geopotential height, from which the SLW path is derived for comparison with the model output.*

*Hersbach, H., Bell, B., Berrisford, P., Hirahara, S., Horányi, A., Muñoz‑Sabater, J., ... & Thépaut, J. N.: The ERA5 global reanalysis. Q. J. R. Meteorol. Soc., 146(730), 1999-2049, https://doi.org/10.1002/qj.3803, 2020.*

• Line 368: I would perhaps simply add that the models lack in reproducing a bimodal SLW distribution.

We have revised this sentence as follows:

"*However, all three experiments fail to reproduce the bimodal distribution of MVD or the first peak near 4 μm, highlighting a common limitation in representing small droplet populations.*"

**Technical comments/questions:**

• line 62: "simply" to "simple"

Corrected

• line 87: "climatic" to "climatological"

Corrected

• line 108: "is" to "were"

Corrected

• Figure 1: The arrow of the flight path is hardly visible. Maybe place it better directly next to flight path or enlarge it a bit. In addition, it would be nice, if you could also plot the innermost nest in panel b. Then it is easier to compare.

Corrected

[Figure]

• Figure 2: The orange line looks more reddish. May name it red instead of orange.

Corrected

• Figure 4: I would recommend collecting the explanation of the isolines in one place in the caption. Explain the part: cloud water: g kg$^{-1}$, rain: 10$^{-3}$ g kg$^{-1}$, cloud ice: 10$^{-7}$ g kg$^{-1}$, snow: 10$^{-3}$ g kg$^{-1}$, temperature: °C together with the shading.

Corrected

[Figure]

• Table 2: Please avoid line breaks in the units (row 1 and 5).

Corrected

• Figure 5: Please provide information in the caption, what QCten is. All other classes can be inferred by the name itself.

Corrected

[Figure]

• line 361: Please exchange "confirm" with "show" or similar. Because you cannot confirm the same thing with the same data. Both plots show this feature in MVD from the same CDP data.

Corrected

• Line 397: Which blue dot are you referring to in Figure 10c. I just see only multiple blue plots from simulation, but not a specific one.

We have revised this sentence as follows:

"*This suggests that CAMS provides a better simulation for larger droplets, as indicated by the cluster of blue dots in Fig. 10c, which coincide with the observed second peak of MVD.*"